# The OASIS walking study—Older adults with cognitive impairment performing sit to stands and walking in transitional care programs: Protocol for a feasibility study

**Alexia Cumal** [1,2], **Tracey J. F. Colella**[1], **Martine T. Puts**[2], **Katherine S. McGilton** [1,2]*

**1** KITE Research Institute, Toronto Rehabilitation Institute, University Health Network, Toronto, Ontario, Canada, **2** Lawrence S. Bloomberg Faculty of Nursing, University of Toronto, Toronto, Ontario, Canada

* kathy.mcgilton@utoronto.ca

**Data Availability Statement:** The participants of this study are not providing written consent for their data to be shared publicly, so due to the sensitive nature of the research, supporting data is

## Abstract

Older adults with cognitive impairment often experience low mobility and functional decline in hospital, transfer to facility-based transitional care programs, and have poorer outcomes compared to those without cognitive impairment. This protocol paper describes a study which aims to determine the feasibility of, satisfaction with, and efficacy of a nurse-led mobility intervention (OASIS Walking Intervention) for older adults with cognitive impairment in facility-based transitional care programs in Ontario, Canada. A quasi-experimental one-group time series feasibility study will be conducted. A sample size of 26 participants will be recruited from two transitional care programs in Ontario, Canada. Participants will receive the OASIS Walking Intervention for up to 45 minutes per session, 5 sessions per week, for 6 weeks. The intervention consists of: 1) a patient-centered communication care plan; 2) sit to stand activity; and 3) a walking program. Feasibility will be determined by: a) recruitment rate; b) retention rate; and c) adherence. Efficacy of the intervention will be determined by the change over time in older adults' lower extremity muscle strength, mobility, and functional status and by their discharge destination (home vs. nursing home). Satisfaction will be measured using the Client Satisfaction Questionnaire. Efficacy outcomes will be measured before the start of the intervention, after 3 weeks of the intervention, and immediately after 6-week intervention. Descriptive statistics will be used for measures of feasibility, satisfaction, and discharge destination. Repeated measures analysis of variance (RM-ANOVA) will be used to analyze efficacy. Ethics approval has been received for this study. Findings from the study will be used to refine the intervention for use in a definitive pilot trial. Results will be disseminated via peer-reviewed publications, international conferences, through group presentations at the study sites, and through the study site networks.

**Trial registration**: The trial has been registered on Clinicaltrials.gov (NCT06150339).

## Introduction

Cognitive impairment (CI), which can include dementia, delirium, and unspecified CI [1, 2], has a global prevalence of 5.1% to 41%, and a median of 19.0% [3]. For dementia in particular,

not available. The Research Ethics Committee in accordance with the Tri-Council Statement in Canada has not given permission to share the data. Due to the nature of the interviews, the data cannot be de-identified further to be able to share them anonymously. The contact information of the University Health Network is reb@uhnresearch.ca contact info+ 1 416-581-7849.

**Funding:** Funding This work is supported by a Doctoral Research Award: Canada Graduate Scholarships from the Canadian Institutes of Health Research grant number [202111FBD-476735-DRA-ADHD-96200] (https://cihr-irsc.gc.ca/e/193.html) and by funding from the Lawrence S. Bloomberg Faculty of Nursing at the University of Toronto (https://bloomberg.nursing.utoronto.ca/) for A.C. K.S.M. is supported by the Walter & Maria Schroeder Institute for Brain Innovation and Recovery (https://schroederfoundation.org/). The funders had no role in the study design, data collection and analysis, decision to publish, or preparation of the manuscript.

**Competing interests:** The authors have declared that no competing interests exist.

the number of people living with this condition globally is expected to nearly double from 50 million in the year 2020 to 82 million in 2030, and 152 million in 2050 [4]. In Canada, the number is expected to nearly triple from about 600,000 in 2020 to 1.7 million in 2050 [5].

Older adults (≥65 years) [6] with CI are frequently hospitalized [7]. While in hospital, they often experience mobility limitations [8], low activity levels [9], and hospital-associated functional decline (HAFD) [10], that is, the inability to perform usual activities of daily living (ADLs) due to weakness, reduced muscle strength, and reduced exercise capacity, which occurs due to bedrest, deconditioning, and acute illness during hospitalization [11].

Older adults often experience HAFD [11–13], with a recent meta-analysis reporting a 30% prevalence of HAFD among hospitalized older adults [13]. This is problematic as HAFD can have serious consequences for the health of older adults with CI. HAFD can lead to not only decreased mood and quality of life [14], but also to pressure injuries, falls, increased morbidity, and mortality [15]. Moreover, HAFD can lead to rehospitalization, increased health care costs, and institutionalization [15]. Many older adults with CI who experience HAFD are subsequently transferred to facility-based transitional care programs (TCPs), which are short-term, post-acute care facilities that provide low-intensity restorative care [6]. In Ontario, facility-based TCPs have been created for patients with a prolonged hospital length of stay who are labelled as Alternate Level of Care (ALC) and are unable to be discharged home; many are waiting to be discharged to a nursing home post hospitalization [16] and many experience a decline in their ability to perform ADLs [17]. However, in a systematic review led by the first author [18], it was found that older adults with CI in facility-based TCPs experience worse outcomes than those without CI. While those with CI had improvements in functional status in eight of 13 studies, a greater percentage of participants without CI experienced higher functional improvement and gains in functional status were smaller for older adults with CI compared to those without CI [18]. Moreover, a smaller percentage of older adults with CI were discharged home post TCP, compared to those without CI [18]. Given the growing number of older adults with CI, there is an urgent need for additional supports and interventions in the TCP setting to improve outcomes for this population.

HAFD can be explained using the pathophysiology of functional decline as described by the Cascade to Dependency [19] and functional decline secondary to muscle disuse models [20]. Together, these models explain that aging-related factors (reduced muscle strength and aerobic capacity) combined with hazards of hospitalization (such as bedrails and tethers that promote immobility, inactivity, and bedrest) result in disuse-induced functional decline [19, 20]. Disuse-induced functional decline is characterized by muscle atrophy, loss of muscle strength, and functional deterioration, all of which increase the risk for admission to a nursing home [19, 20]. To counteract the deconditioning and functional decline, there is a need for anabolic strategies. which build muscle, such as walking and resistance training which promote muscle growth [20]. Resistance training such as rising from a seated chair position to standing as fast as possible, using one's body weight as resistance, can greatly increase muscle mass and strength and improve functional ability [20].

## Literature review

A comprehensive search of the literature involving mobility interventions led by nurses yielded: zero studies in the TCP setting; six studies with significant results involving older adults with cognitive impairment and walking interventions that were led by nurses or that could be done by nurses in the nursing home setting [21–28]; and one study with significant results involving sit to stand activity in the nursing home setting [29]. No studies were found that combined these two interventions, mobility and sit to stand.

In nursing homes, walking had a typical dose of 30 minutes per session [22, 23, 26–28], ranged from two [21] to seven [25] sessions per week; and duration ranged from six weeks [26] to six months [27]. Overall, the most effective intervention was administered by a nurse and involved 2–4 walking sessions per week for 4 months and incorporated a person-centred communication care plan [21]. This study had the highest recruitment and adherence rates, monitored intervention fidelity, and resulted in a significant improvement to all outcomes (functional mobility, activities of daily living function, and quality of life). One study involving older adults with dementia from seven nursing homes found that doing six sit to stands per day for six months resulted in maintained mobility (measured by the amount of time to perform one sit-to-stand) (p = 0.01), which can also be a measure of lower extremity muscle strength [29].

## Gap in the literature

To the authors' knowledge, there have been no nurse-led intervention studies completed to date which combine sit to stand activity, a walking intervention, and a patient-centred communication care plan for older adults with CI in facility-based TCPs. In order to address this gap in the literature, a feasibility study needs to be undertaken. A feasibility study is designed to assess an intervention, including optimal content, delivery, and adherence to the intervention, as stated in the new framework on complex intervention research that was commissioned by the Medical Research Council (MRC) and the National Institute of Health Research (NIHR) [30]. This framework outlines that a feasibility study should be done to test the feasibility of a complex intervention in order to make decisions about progression to the next stage of evaluation [30].

## Aim

The aim of this study is to determine the feasibility of and participant satisfaction with a novel intervention–the Older Adults with cognitive impairment performing Sit to Stands and Walking Intervention (that is, the Older Adults with cognitive impairment performing Sit to Stands and Walking Intervention) in facility-based TCPs. The second aim is to determine the efficacy of the OASIS Walking intervention on muscle strength, mobility, functional status, quality of life, and discharge destination.

## Research questions

1. What is the feasibility of implementing the OASIS Walking Intervention in community-dwelling older adults with CI in facility-based TCPs, as determined by recruitment rate, retention rate, and adherence?

2. What is the satisfaction of older adults with CI with the OASIS Walking Intervention?

3. Does the OASIS Walking Intervention result in improved muscle strength, mobility, functional status, and quality of life in older adults with CI?

4. What percentage of the participants are discharged home and how many are discharged to the nursing home post intervention?

## Study hypotheses

- It is hypothesized that the OASIS Walking Intervention will be feasible, defined as a recruitment rate (>50%), high retention rate (≥80%), and high adherence rate (attendance to ≥75% of all intervention sessions) [26].

- It is hypothesized that the participant satisfaction will be high (CSQ of 3 or more on all SCQ items) [31].

- It is hypothesized that the OASIS Walking Intervention will result in improved muscle strength (reduction in the time to perform one sit to stand by ≥0.87 seconds, p = 0.01 [29], mobility (minimum detectable change (MDC) ≥9.1 meters on two-minute walk test [32]), functional ability (≥1 point improvement in Barthel Index (BI) considered as clinically meaningful [33]) in older adults with CI in facility-based TCPs, and quality of life (minimally clinical important difference is an increase in 3 points [34]).

## Methods and analysis

### Study design

A feasibility study will be undertaken for this three-component intervention project. A feasibility study is in keeping with the MRC and NIHR framework which states that for complex interventions, a feasibility study is done to assess and refine the intervention prior to carrying out a full-scale evaluation [30]. In terms of study design, a quasi-experimental single group time series design will be used.

The trial was registered on Clinicaltrials.gov (NCT06150339) on November 29, 2023.

### Setting

The study will take place in two Transitional Care Units (TCUs) in Ontario, Canada, one in Pickering and one in Scarborough. The units have a combined total of 107-bed capacity. From January 2024-June 2024, patients will be enrolled in the study on an ongoing basis. Up to 8 patients will be enrolled in the study at a given time.

### Participants and recruitment

Older adult patients will be eligible if they meet the following criteria: 1) aged 65 years and older; 2) have cognitive impairment (dementia, delirium, mild cognitive impairment, or unspecified cognitive impairment) as documented in the medical record or Quick Dementia Rating Scale (QDRS) score of ≥2) (S1 Appendix); 3) admitted to a transitional care unit after a hospitalization; 4) can speak English; 5) has received clearance from the physiotherapist to participate in the study; 6) has received clearance from the nurse practitioner to participate in the study; 7) were community-dwelling (lived in a home or retirement home; not a nursing home) prior to hospitalization; 8) were able to ambulate independently or with the assistance of one person (with or without a gait aid) prior to hospitalization; 9) is currently able to ambulate either independently or with the assistance of one person (with or without a gait aid); 10) has a care partner (family member, friend) who is willing participate in an interview about the patient for the study.

A diagnosis of CI, such as dementia is often under-reported in clinical records [35], and so the second inclusion criterion of have CI (dementia, delirium, mild cognitive impairment, or unspecified CI) will be ascertained in one of two ways: 1) a diagnosis of CI in the medical record; or 2) QDRS score of ≥2. The QDRS is a rapid (3–5 minutes) 10-item questionnaire that is used in clinical research to assess the presence of CI for inclusion and inclusion into studies [36]. The questionnaire is informant-based; informants can be spouses, adult children, relatives and friends, and paid caregivers [36]. For this study, the informant will be the substitute decision maker (SDM) of the participant and the QDRS will be evaluated by the interventionist (AC), who is a Registered Nurse. A score of 0–1 indicates a high likelihood of

normal cognition; 2–5 indicates mild cognitive impairment, 6–12 indicates mild dementia; 13–20 indicates moderate dementia; and 20–30 indicates severe dementia [37]. The QDRS has good reliability (Cronbach α 0.86–0.93), demonstrates similar validity as the longer clinical dementia rating (CDR) scale to detect the presence of CI in older adults [36].

### Exclusion criteria

Patients will be excluded if they are: 1) Palliative (having <six months prognosis as defined by Hui and colleagues [38]) as documented in the medical chart; 2) have Parkinson's disease as documented in the medical chart (due to impairments in muscle and motor function) [21].

### Recruitment

To achieve adequate participant enrolment, staff have been provided with detailed information regarding eligibility of patients for the study. Staff at the TCUs will screen participants for eligibility into the study. If the patients meet eligibility criteria, the staff will inform the interventionist (AC) who will approach patients for interest in the study. Informed consent (S2 and S3 Appendix) will be obtained from patients who pass the Evaluation to Sign Consent (ESC) Measure (S4 Appendix) [39]. For patients who do not pass the ESC but who assent to the study, informed consent will be obtained from their substitute decision maker (SDM). The SPIRIT Schedule of enrolment, interventions, and assessments for this study is outlined in S1 Fig.

### Sample size

An activity-based study involving older adults with dementia in the nursing home setting [29] that used time to perform one sit to stand found a moderate effect size (Cohen's d) of 0.48. Since the present study will use repeated measures ANOVA, a Cohen's $f$ of 0.25 (a suggested value for moderate effect size) [40] was used in the sample size calculation. Based on an $f$ of 0.25, a power of 0.8 and an alpha of 0.1, in keeping with those used in a mobility study involving older adults with dementia [21], a sample of 21 participants will be needed for this study. Using an alpha of 0.1 can be acceptable for exploratory or preliminary studies [41] (p. 188). An attrition rate of just under 20% over the course of a 6-week study will be taken into account, as was experienced in a 6-week intervention study involving older adult residents with dementia [26]. Thus, a sample of 26 participants will be recruited for this study. G*Power 3.1.9.7 software [42] was used for the sample size calculation for an ANOVA: Repeated measures, within factors statistical test with one group, three measurements, 0.6 for the correlation among repeated measures, and 0.8 for nonsphericity correction (which is a mild departure from sphericity) [41].

### Intervention

The OASIS Walking Intervention is a nurse-led intervention that consists of three components: 1) Patient-Centred Communication Care Plan; 2) Sit to Stand Activity; and 3) Walking program. The interventionist is a Master's prepared registered nurse with 10 years of clinical nursing experience on a General Internal Medicine Unit in an urban acute care hospital working with older adults with cognitive impairment, including walking, transferring, and communicating with these patients. The interventionist and research assistants will have received additional training from the unit physiotherapist on transfer training, walking, and performing the two-minute walk test. The intervention is grounded using a patient centred approach. An intervention manual has been created for this study (S5 Appendix).

## Intervention dose

The dose of the intervention is: up to 45 minutes per session, five sessions per week, for six weeks. Approximately up to 30 minutes will be spent walking with the participant and up to 15 minutes will be spent performing the sit-to-stand activity as per their tolerance levels.

This intervention goes beyond usual care provided the TCUs. There is no change to usual care provided to patients as a result of participating in this intervention. For patients in the long-term care stream usual care consists of: one-to-one sessions 2–3 times per week of strengthening and balance exercises. For patients in the Rehab stream, usual care consists of: one-to-one sessions 5 times per week of strengthening, balance, and may include some walking and group therapy 5 times per week.

## Intervention components

**Component 1.** Patient-Centred Communication Care Plan. This care plan will be informed by interviews that the interventionist will have with the participant and their care partner. During the interviews, the interventionist will ask questions about three areas of a patient-centred assessment: 1) Participant's biography (work history, family, interests); 2) Participant's communication abilities and preferences; 3) Engagement with the Participant; [21, 43]. The information gained from the interviews will be added to the patient-centred communication template (S2 Fig). The individualized care plan will be used during the walking and sit to stand activity sessions to promote enjoyment, engagement, adherence, and communication.

**Component 2.** Sit to Stand Activity. The procedure for the sit to stand activity, which is adapted from the sit to stand protocol used by Barreca 2004 and colleagues [44], is outlined in Table A, which is embedded in the intervention manual (S5 Appendix).

Target Number of Sit to Stands. The interventionist will measure the number of sit to stands that a participant can do in 30 seconds at baseline, as used in previous study [29]. Based on that number, a target number of sit to stands to do per session will be determined, based on the algorithm used in the study by Slaughter and colleagues [29]). Halfway into the intervention (after 3 weeks), the target number of sit to stands will be progressed. The target number of sit to stands and the algorithm used by Slaughter et al. [29] (Personal communication with S. Slaughter January 2023) was adapted into the formula for this study:

Weeks 1–3:

Target number of sit to stands per session
$$= \frac{1}{2} \times [4 \times (\textit{Number of sit to stands done in } 30 \textit{ seconds} + 1)]$$

Weeks 4–6:

Target number of sit to stands per session
$$= 4 \times (\textit{Number of sit to stands done in } 30 \textit{ seconds} + 1)]$$

Table 1 outlines the calculated target numbers of sit to stands for the present study.

**Table 1. Target numbers of sit to stands.**

| Number of Sit to Stands done in 30 seconds at Baseline (n) | Initial Target Number of Sit to Stands per Intervention Session(Weeks 1–3) | Progressed Target Number of Sit to Stands per Intervention Session (Week 4–6) |
|---|---|---|
| 1 | 4 | 8 |
| 2 | 6 | 12 |
| 3 | 8 | 16 |
| 4 | 10 | 20 |
| 5 | 12 | 24 |
| n | $= \frac{1}{2} \times [4 \times (n+1)]$ | $4 \times (n+1)$ |

**Component 3.** Walking Program. Based on the findings from the patient-centred assessment interviews as well as the performance of the participants on the two-minute walk test at baseline (Time 1), an individualized walking program will be carried out with participants, in a manner similar to the process done in the study by Chu and colleagues [21]. The goal will be to walk up to 30 minutes each session, five days per week. The interventionist will personalize the dose duration and speed of each walking session as tolerated by the participant [21] and as assessed using the Borg Rate of Perceived Exertion (RPE) scale [45, 46].

## Outcome measures

### Data collection

Demographic variables, including age, sex, gender, race/ethnicity, education, socioeconomic status, highest level of education, and number of co-morbidities will be collected before the start of the intervention (S6 Appendix). Data on hospital admission diagnosis and length of hospital stay prior to TCP admission will also be collected (S6 Appendix). Table 2 outlines the outcome measurement tools that will be used to assess the participants, time points at which they will be measured, and the type of statistical test that will be used to analyze the results.

### Sex and gender considerations during data collection

Participants' sex (defined as a set of biological attributes; sex is usually categorized as female or male [47]) and gender (defined as socially constructed roles, behaviours, expressions, and

**Table 2. Outcome measurement tools, time points, and statistical tests.**

| Research Question Number | Outcome (Tool) | Pretest Initial Assess-ment (T1) | 1 | 2 | 3 | Assess-ment (T2) | 4 | 5 | 6 | Posttest (Immediately after 6-week intervention) (T3) | Type of Statistical Test |
|---|---|---|---|---|---|---|---|---|---|---|---|
| | | | | | | **Weeks of Intervention** | | | | | |
| 1 | Recruitment Rate (documentation) | √ | | | | | | | | | Descriptive statistics |
| | Retention Rate (documentation) | √ | √ | √ | √ | √ | √ | √ | √ | √ | Descriptive statistics |
| | Adherence (checklist) | | √ | √ | √ | √ | √ | √ | √ | | Descriptive statistics |
| | Intervention Fidelity (checklist) | | √ | √ | √ | | √ | √ | √ | | Descriptive statistics |
| 2 | Satisfaction (Client Satisfaction Questionnaire + 3 open ended questions) | | | | | | | | | √ | Descriptive statistics; Content analysis |
| 3 | Lower extremity muscle strength (Time to perform one sit to stand) | √ | | | | √ | | | | √ | Means and SDs, repeated measures ANOVA |
| | Mobility (2-minute walk test) | √ | | | | √ | | | | √ | Means and SDs, repeated measures ANOVA |
| | Functional Status (Barthel Index) | √ | | | | √ | | | | √ | Means and SDs, repeated measures ANOVA |
| | Quality of Life (scale) | √ | | | | √ | | | | √ | Means and SDs, repeated measures ANOVA |
| 4 | Discharge Destination (Chart review) | | | | | | | | | √ (At the time of discharge from the TCU or within 60 days of admission to TCU, whichever comes first) | Descriptive statistics |

T1 = time 1; T2 = time 2; T3 = time 3; ANOVA = analysis of variance; SDs = standard deviations.

identities of women, men, and gender diverse people; gender is usually conceptualized as a binary (woman and man) yet there is considerable diversity in how individuals understand, experience, and express it [48]. Sex and gender will be assessed in the demographic questionnaire (S6 Appendix).

## Measures

**Primary outcome.**  *Feasibility*. Feasibility will be measured through recruitment rate, retention rate, and adherence [49].

1.  *Recruitment rate*. Recruitment rate will be calculated as the percentage of participants who enroll in the study out of the total number of eligible participants [21]. Reasons for nonenrolment and ineligibility for the study will also be recorded. A recruitment rate of >50% is considered moderate [26].

2.  *Retention rate*. Retention rate will be calculated as the percentage of participants who complete the study (i.e., receive the full dose of the intervention and provided post-test outcome data) out of the number of participants who were enrolled (i.e., signed a consent form and provided baseline data) [49]. A retention rate of ≥80% is considered high [26].

3.  *Adherence*. Participants' adherence will be determined by: 1) the average number of treatment sessions attended; and 2) the level of engagement with the treatment [49], that is, a) the duration of each walking session, duration of each intervention session, and number of sit to stands done per session; and 2) b) the number of sit to stands done per session, divided by the goal number of sit to stands. An adherence of ≥75% is considered high [26]. The adherence checklist can be found in S7 Appendix.

**Secondary outcomes.**  Efficacy will be assessed using measures for muscle strength, mobility, functional status, quality of life, and discharge destination. The primary outcome for efficacy will be lower extremity **muscle strength**.

1.  *Lower extremity muscle strength*. Lower extremity muscle strength will be measured using the time to perform one sit-to-stand [50, 51]. Time to perform one sit-to-stand has good validity and reliability. The ability to perform one sit to stand has been used in a study involving sit-to-stand activity with older adults with dementia in nursing homes [29]. Repeated observations of the sit to stand test by one observer yielded correlations of 0.89 to 0.96 [52].

2.  *Mobility*. Mobility will be measured using the two-minute walk test (2MWT). The testing procedure for the 2MWT to be used will be similar to those used in a studies involving older adults with cognitive impairment in nursing homes [32, 53]. The 2MWT has been used in studies involving older adults with dementia in nursing homes [21, 32] and has a test-retest reliability coefficient of 0.98 (0.96–0.99) and an inter-rater reliability of 0.92 (0.86–0.96) when assessed in frail older adults with dementia [32]. The minimal detectable change (MDC), that is, the minimum change that is considered a true change in performance, for the 2MWT is 9.1 metres [32].

3.  *Functional status*. Functional status will be measured by the Barthel Index (BI), which has been shown to have a good reliability (kappa > 0.75) and validity for short stay older adult patients [54] and has acceptable item reliability (1.0) and person reliability (0.88) in older adults with dementia [55]. BI is a 10-item questionnaire which is scored out of 20, with higher scores meaning a better outcome. A change in BI score of 1 point is considered a meaningful change in a person's level of independence [33]. When the BI score is

multiplied by 5, a score of 0–20 indicates total dependency, 21–60 indicates severe dependency, 61–90 indicates moderate dependency, and 91–99 indicates slight dependency [56]. BI scores will be measured at pretest, time 2, and post-test.

4. *Quality of life*. Quality of life will be measured using the QOL-AD Quality of Life–Alzheimer's Disease, a 13-item questionnaire that asks about physical health, energy, mood and other quality of life measures that can be answered by older adults with CI [57]. The QOL-AD has good reliability ($\alpha$ from 0.83 to 0.90) and validity correlation with measures of depression ($r = -0.41$ to $-0.65$) [57]. The QOL-AD will be completed by the SDM if the patient is not able to complete it.

5. *Satisfaction*. Participant satisfaction will be determined using the Client Satisfaction Questionnaire (CSQ) [58], modified to the TCP setting, which includes 8 Likert scale questions, a comments section; and three open ended questions, similar to those used in the work by Sano and colleagues [31]. The CSQ will be completed by the SDM if the patient is not able to complete it. As well, qualitative data in the form of field notes will also be taken on interventionists' report of challenges, issues, and ease of intervention delivery [49]. Field notes taken within the intervention fidelity checklist will also capture interventionists' perspectives on elements of the intervention that are satisfactory to participants and their care partners. The CSQ will be completed by the SDM if the patient is not able to complete it (see S8 Appendix for Time 3 outcome measures with the CSQ and 3 open-ended questions).

6. *Discharge destination*. Planned and actual discharge destinations will be determined by the interventionist through a chart review or confirmation with TCU staff.

**Intervention fidelity.**   Intervention fidelity will also be measured through the interventionist's self-report [49] of 12 intervention items. The percentage will be calculated as the number of items done divided by the 12 items on the intervention fidelity checklist. Any safety events (such as falls) that occur during the intervention sessions will also be documented.

## Additional data collected

At Time 3, the RA will review the participant's chart to document services provided to the participants in addition to the OASIS Walking Intervention, to increase the internal validity of the study. A shortened version of a checklist of services/treatments related to mobility (Table 3) provided to patients in TCPs developed by McGilton and colleagues based on their scoping review [6] will be completed for this study.

Table 3. Checklist of mobility-related services provided.

| Services | Yes | No | n/a | Comment (specify dose where appropriate) |
|---|---|---|---|---|
| Mobility | | | | Daily/weekly/bi-weekly/other: please specify |
| Rehabilitation training including transfers, stairs, strength and balance exercises and provision of mobility aids | | | | Daily/weekly/bi-weekly/other: please specify |
| Functional training in ADL (toileting, washing, dressing, walking) | | | | Daily/weekly/bi-weekly/other: please specify |
| Psychosocial care measures such as central dining, recreational activities, group exercises, spiritual care | | | | Daily/weekly/bi-weekly/other: please specify |

## Statistical analysis

**Research question 1.** What is the feasibility of implementing the OASIS Walking Intervention in community-dwelling older adults with CI in facility-based TCPs, as determined by recruitment rate, retention rate, and adherence?

Data will be analyzed using SPSS version 28.0.1.0. Descriptive statistics will be used for the demographic variables as well as for measures of feasibility (recruitment rate, retention rate, and adherence). Specifically, for measures of adherence, the mean number of treatment sessions attended will be calculated and the range of treatment sessions will be provided. In terms of level of engagement with the treatment, the mean duration of each walking session, of each intervention session, and number of sit to stands done per session will be calculated. As well, the range of each of these durations will be provided. Furthermore, the mean number of sit to stands done per session, divided by the goal number of sit to stands will be calculated.

**Research question 2.** What is the satisfaction of older adults with CI with the OASIS Walking Intervention?

Satisfaction will be also assessed through the opened ended questions in the CSQ satisfaction surveys. Quantitative data will be analyzed using descriptive statistics; qualitative data will be analyzed using content analysis [59].

**Research question 3.** Does the OASIS Walking intervention result in improved muscle strength, mobility, functional status, and quality of life in older adults with cognitive impairment?

Means and standard deviations for each of the outcome measures will be summarized. Longitudinal plots of overlaid individual trajectories will be used to visualize observed change over time. Repeated measures analysis of variance (ANOVA) [60, 61] will be used to determine if the intervention results in an improvement in participants' time to perform one sit to stand, 2MWT, BI, and quality of life over time. Repeated measures ANOVA is a parametric test that determines if the means of three or more measures from the same person are similar or different [61]. Scatter plots will be used to demonstrate any changes in outcomes. If there is missing data, paired t-tests will be used to determine if there is an improvement in participants' time to perform one sit to stand, 2MWT, BI, and quality of life between two time points.

As well, simple tests of before–after (paired t-tests) will be used to determine the differences between Time 1 (baseline) and Time 2 (after 3 weeks of the intervention), and between Time 1 and Time 3 (after the 6-week intervention) for the outcome variables of interest. These tests will yield corresponding 95% confidence intervals, allowing investigators a sense of the magnitude of changes that one might look for in any subsequent trial.

**Research question 4.** What percentage of the participants were discharged home and how many were discharged to the nursing home post intervention? Percentages will be used to describe discharge destination of participants.

## Sex and gender considerations in analysis

Descriptive statistics will be used to report on sex and gender. Specifically, descriptive statistics will be used to report data that is disaggregated by sex and gender.

## Criteria to evaluate feasibility and potential for success of a future definitive trial

Feasibility of this study is being evaluated by recruitment rate, retention rate, and adherence rate. If the study has a recruitment rate of >50%, a retention rate of ≥80%, and an adherence rate of ≥75%, these will demonstrate feasibility of the study, and therefore potential for success

in a future definitive trials. Moreover, if there is high satisfaction with the intervention (CSQ of 3 ore more on the CSQ-8 Questionnaire), this will provide additional evidence to support the testing of this intervention in a more definitive trial.

## Access to source documents and confidentiality

Data will only be accessed by the research staff for the purpose of the study. These individuals will complete privacy training and signed confidentiality agreements and/or will be required by law to keep all collected information confidential. Representatives of the University Health Network (UHN) including the UHN Research Ethics Board may be given remote access to an electronic portal (via the internet) to look at the study records to check that the information collected for the study is correct and to make sure the study is following proper laws and guidelines. The electronic data will be kept in a secure one drive storage database hosted by UHN, which has restricted access and safety backup.

All personal information such as participants' name will be removed from the data and will be replaced with a number. A list linking the number with participants' names will be kept by the study investigator in a secure place, separate from participant files. Whether on-site or remotely, UHN makes all efforts to ensure that participant information is shared in a way that is secure and private (encrypted). The research team will keep any personal health information about participants in a secure and confidential location for 10 years.

## Procedures for data security

The research team will engage in systematic data management and adhere to high standards to protect participants' confidentiality. Study data will be protected using several strategies. Participants will be given a unique identifier number. This code will not have anything to do with participants' names. Physical copies of data will be stored in a locked cabinet at the PI's office. Participants' contact information, study assigned ID, signed consent forms will be stored securely and separately from completed data collection records. All data will be stored on the TRI–UHN server. The interview with participants and care partners at the beginning of the study will be audiorecorded and then transcribed. All transcripts will be anonymized, and audio recordings will be destroyed upon transcription. Upon completion of the study, data will be archived in a secure, locked location for ten years, then destroyed. In the event of inappropriate release of data, all attempts will be made to stop further release, and any information that could be retrieved will be retrieved. The UHN Privacy Office will be notified, and further actions will be taken according to the UHN Privacy Office and REB recommendations.

## Ethics statement and dissemination

The study has received written ethical approval from the University Health Network Research Ethics Board (Study ID 23–5543). Specifically, the approval letter stated: "The University Health Network Research Ethics Board approves the above mentioned study as it has been found to comply with relevant research ethics guidelines, as well as the Ontario Personal Health Information Protection Act (PHIPA), 2004" (p. 2). Should there be any important protocol modifications, the relevant parties (research ethics boards, investigators, trial participants, trial registry) will be notified. Informed written consent will be received from each participant in the study.

If participants are harmed as a direct result of taking part in this study, all necessary medical treatment will be made available to them at no cost.

Findings from the study will be used to refine the intervention for use in a definitive pilot trial. Results will be disseminated via peer-reviewed publications, international conferences, through group presentations at the study sites, and through the study site networks.

The study has been registered on Clinicaltrials.gov (NCT06150339). The study details can be found at https://clinicaltrials.gov/study/NCT06150339.

## Remuneration

Participants will not have to pay for any procedures involved in this study. As a token of appreciation and in recognition of their time and effort, a $5 gift card to a coffee shop will be given to participants at each of the three outcome measurement stages of the research. By the end of the study, participants will be given a total of $15 in gift cards. A $10 gift card to a coffee shop will be given to care partners of participants after their interview as a token of appreciation for their time. Providing a token gift at each stage of the research follows the guidance provided by the Division of the Vice President, Research & Innovation at the University of Toronto [62]. Moreover, a recent Cochrane systematic review by Gillies and colleagues found that monetary incentives can increase participant retention in intervention studies [63].

## Plan for missing data

A plan to minimize the risk of missing data will be put in place. Specifically, an Ethical Protocol and Algorithm for Data Collection and Intervention Session: Procedure for Assessing Assent and Dissent for the OASIS Walking Intervention (S9 Appendix) will be used. This protocol is adapted from the Ethic Protocol used by Chu and colleagues [21]. For their study, all follow up assessments were completed and there was no missing data [21]. This protocol includes establishing rapport with participants, obtaining assent prior to initiating data collection, and re-approaching if the participant initially refuses (S9 Appendix). Data on paper sheets will also be visually inspected for missing data prior to transferring them onto the secure One drive. In addition to the above retention strategy, remuneration will also be provided to reduce the risk of drop out and thus reduce the risk of missing data.

## Discussion

This article describes the protocol for a feasibility study of a nurse-led mobility intervention. Hospitalized community-dwelling older adults with CI often experience functional decline and are admitted to facility-based TCPs, where they have poorer outcomes, including functional status and discharge destination [18]. Walking programs in hospitals and long-term care homes have been found to improve outcomes for older adults with CI such as functional status, quality of life, and mobility. As well, programs involving sit to stand activity have shown significant results in terms of lower extremity muscle strength in long-term care homes. However, to the authors' knowledge, no such combined intervention has been done in facility-based TCPs.

This protocol outlines a 6-week nurse-led mobility intervention for older adults with CI in TCPs which incorporates a patient-centred communication care plan, sit to stand activity, and a walking program. Outcome measurements will take place at pretest, after 3 weeks of intervention, and posttest.

The results of this feasibility study will be valuable for optimizing the design of a definitive controlled trial that could impact clinical practice in this population and setting. If the study demonstrates a recruitment rate of >50%, a retention rate of ≥80%, and an adherence rate of ≥75%, these will demonstrate feasibility, and will thus provide evidence to proceed with a definitive controlled trial. Moreover, if the study results in high satisfaction with the intervention, it will provide further evidence for potential for success in a future definitive trials. Furthermore, the open-ended questions, which includes the question, "If I could change one thing about the walking program, it would be:" which will be asked together with the satisfaction

questionnaire will provide additional data which might inform modifications to the design of a trial.

The use of one group rather than having two groups is a limitation of this feasibility study. Still, this study will provide evidence on the feasibility and efficacy of a nurse-led mobility intervention in TCPs. It will provide valuable insight on the feasibility of the components of the intervention and on patients' satisfaction with the intervention. Furthermore, it will provide preliminary evidence which can inform a definitive pilot study.

## Conclusion

A six-week nurse-led mobility intervention which aims to improve functional status, mobility, quality of life, and satisfaction among older adults with CI in TCPs was described. This study will provide valuable insight into the feasibility of this intervention in this setting. Study findings will allow for the refinement of intervention components for a full-scale pilot study.

## Supporting information

**S1 Text. SPIRIT checklist.**
(DOC)

**S1 Fig. SPIRIT schedule of enrollment, interventions, and assessments.**
(TIF)

**S2 Fig. Patient-centred communication care plan.**
(TIF)

**S1 Appendix. Quick dementia rating scale.**
(PDF)

**S2 Appendix. Consent form–participant + SDM.**
(DOCX)

**S3 Appendix. Consent form–care partner.**
(DOCX)

**S4 Appendix. Evaluation to sign consent.**
(DOCX)

**S5 Appendix. Intervention manual.**
(DOCX)

**S6 Appendix. Time 1 assessment.**
(PDF)

**S7 Appendix. Adherence checklist.**
(DOCX)

**S8 Appendix. Time 3 assessment with CSQ-8 and 3 open ended questions.**
(PDF)

**S9 Appendix. Assent and ethical protocol.**
(DOCX)

**S10 Appendix. Protocol for OASIS walking study that was approved by ethics committee.**
(DOCX)

## Author Contributions

**Conceptualization:** Alexia Cumal, Tracey J. F. Colella, Martine T. Puts, Katherine S. McGilton.

**Funding acquisition:** Alexia Cumal.

**Methodology:** Alexia Cumal.

**Supervision:** Tracey J. F. Colella, Martine T. Puts, Katherine S. McGilton.

**Writing – original draft:** Alexia Cumal.

**Writing – review & editing:** Alexia Cumal, Tracey J. F. Colella, Martine T. Puts, Katherine S. McGilton.

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
