## [Decision Letter · Decision Letter 0]

15 May 2024

PONE-D-24-00334The OASIS Walking Study - older adults with cognitive impairment performing sit to stands and walking in transitional care programs: Protocol for a feasibility studyPLOS ONE

Dear Dr. McGilton,

Thank you for submitting your manuscript to PLOS ONE. After careful consideration, we feel that it has merit but does not fully meet PLOS ONE’s publication criteria as it currently stands. Therefore, we invite you to submit a revised version of the manuscript that addresses the points raised during the review process.

We look forward to receiving your revised manuscript.

Kind regards,

Mario Ulises Pérez-Zepeda, M.D., Ph.D.

Academic Editor

PLOS ONE

Journal Requirements:

2. In the online submission form, you indicated that individual participant data that underlie the results that come from the study in this article, after deidentification (text, tables, figures, and appendices) will be available upon request beginning 9 months and ending 36 months following results article publication to Investigators whose proposed use of the data has been approved by an independent review committee ("learned intermediary") identified for this purpose for the purpose of individual participant data meta-analysis.. 

Reviewers' comments:

Reviewer's Responses to Questions

**Comments to the Author**

1. Does the manuscript provide a valid rationale for the proposed study, with clearly identified and justified research questions?

Reviewer #1: Yes

Reviewer #2: Yes

Reviewer #3: Yes

2. Is the protocol technically sound and planned in a manner that will lead to a meaningful outcome and allow testing the stated hypotheses?

Reviewer #1: Partly

Reviewer #2: Yes

Reviewer #3: Partly

3. Is the methodology feasible and described in sufficient detail to allow the work to be replicable?

Reviewer #1: Yes

Reviewer #2: Yes

Reviewer #3: Yes

4. Have the authors described where all data underlying the findings will be made available when the study is complete?

Reviewer #1: Yes

Reviewer #2: Yes

Reviewer #3: Yes

5. Is the manuscript presented in an intelligible fashion and written in standard English?

Reviewer #1: Yes

Reviewer #2: Yes

Reviewer #3: Yes

6. Review Comments to the Author

You may also provide optional suggestions and comments to authors that they might find helpful in planning their study.

Reviewer #1: Major Revision Required

This appears to be a well thought out study but the statistical method sections require extensive review.

The authors describe their design as ‘a quasi-experimental single group time series design’ (page 8, lines 162-3) in which assessments at 0, 3 and 6 weeks will be made (page 2, line 41-2). This can be described as repeated measures design with v = 1 pre- and w = 2 post-intervention assessments. Using the design criteria of 1-sided α = 0.1, power 1 − β = 0.8, δ = 0.48 then using a, v = 1 and w = 1, before-and-after design gives a sample size N = 21 as is stipulated on page 10, line 204. However, with the design chosen, that is v = 1 and w = 2 an explanation of how this becomes compatible with δ = 0.21 is required. This assumption would seem to imply that the change from baseline T1 to T2, that is T2 – T1, will be assumed the same as T3 − T1. In which case, there is an assumed plateau in values between T2 and T3 (see comment on statistical analysis below).

However, it would be very useful to quote the actual methodology used for the sample size calculations not just a general reference to G*Power (page 10, lines 208-9).

It seems very sensible to adjust the sample size to 26 to take account of possible attrition.

In general terms, a feasibility study is not anticipated to establish ‘statistically significant’ differences so that in this case, simple tests of before – after differences at 3 weeks and at 6 weeks for the outcome variables of interest are all that is required with the corresponding 95% confidence intervals (rather than the p-values) of principal interest. These would give the investigators some sense of the magnitude of changes one might look for in any subsequent trial.

Consequently, I suggest that most of the methodology described in Page 20, lines 389-398 will not be appropriate.

Note: Page 13, Table 2: Seems to suggest the T2 assessment is made at 4 not 3weeks -

Reviewer #2: This study protocol paper aims to examine the feasibility and effects of nurse-led older adults with cognitive impairment performing sit-to-stands and walking in transitional care programs. While the study is meticulously planned and serves as a valuable protocol paper, I have provided comments due to several points of inquiry.

1, I'm not well-versed in the healthcare system in Ontario and Transitional Care Programs, so please correct me if I'm mistaken.

1-1, TCPs are provided for individuals who find it difficult to return home from acute care hospitals (due to factors like HADF) and need temporary placement until they can be accommodated in nursing home. Individuals with CI in TCPs are at higher risk of further HAFD, which is associated with adverse outcomes. With this understanding, I believe the sentence of HAFD in lines 77-82 is unnecessary or inserted around line 62.

1-2, Is rehabilitation not provided in TCPs? If rehabilitation is provided, does its participation and frequency not affect OASIS Walking program adherence?

1-3, Are there concerns about facility placement during the intervention period?

2, Regarding subject recruitment

2-1, Please specify the timing of the evaluation of Quick Dementia Rating Scale (QDRS) scores and whether it will be assessed in all older adult patients, as well as add details about the evaluators.

2-2, Is 'delirium' assessed based on it prior to admission to the TCU?

2-3, What does 'cognitive impairment, or unspecified cognitive impairment as documented in the medical record' entail? Does it include a diagnosis of MCI?

2-4, Although adherence to the OASIS Walking programs and its effects may be influenced by the cause of hospitalization, should this not be considered in subject recruitment?"

Reviewer #3: Dear authors,

This protocol addresses an important evidence gap and is well-designed as a feasibility study. Key strengths include a clear rationale, novel intervention components, inclusion of appropriate feasibility and preliminary efficacy measures, and registration on ClinicalTrials.gov.

To further strengthen the manuscript, it is suggested to provide additional details on the qualifications and training of those delivering the intervention, procedures for measuring adherence and participant satisfaction, a statistical analysis plan focused on the feasibility objectives (avoiding emphasis on hypothesis testing of efficacy given the small sample size), and procedures for data security. Reporting these methodological details, even in a feasibility study, will enhance the replicability, interpretation, and usefulness of the findings to inform future trials.

It is also suggested to more explicitly articulate the pre-specified criteria that will be used to evaluate feasibility and potential for success of a future definitive trial, as well as the plan to obtain participant feedback on acceptability of the intervention and study procedures. Finally, it would be valuable to clearly outline in the discussion how the results of this feasibility study will be used to make the decision whether or not to proceed with a definitive controlled trial and how they might inform modifications to the design of such a trial.

Including these details, along with the included trial registration number and an explicit statement about ethical approval, aligns well with the CONSORT extension guidelines for pilot and feasibility trials (available at https://doi.org/10.1186/s40814-016-0105-8). Following these recommendations would strengthen the completeness and transparency of the protocol.

With these revisions, the manuscript will be more robust and ready to inform the important next stage of study implementation. The results of this feasibility study will be valuable for optimizing the design of the eventual definitive controlled trial that could impact clinical practice in this population and setting.

7. PLOS authors have the option to publish the peer review history of their article (what does this mean?). If published, this will include your full peer review and any attached files.

Reviewer #1: No

Reviewer #2: No

Reviewer #3: No

---

## [Author Response · Author response to Decision Letter 0]

2 Jul 2024

Please find attached the document entitled Response to Reviewers, in which we respond to the reviewer and editor comments. Thank you.

---

## [Editor Report · Decision Letter 1]

22 Jul 2024

The OASIS Walking Study - older adults with cognitive impairment performing sit to stands and walking in transitional care programs: Protocol for a feasibility study

PONE-D-24-00334R1

Dear Dr. McGilton,

We’re pleased to inform you that your manuscript has been judged scientifically suitable for publication and will be formally accepted for publication once it meets all outstanding technical requirements.

Kind regards,

Mario Ulises Pérez-Zepeda, M.D., Ph.D.

Academic Editor

PLOS ONE
---

## [Editor Report · Acceptance letter]

13 Aug 2024

PONE-D-24-00334R1 

PLOS ONE

Dear Dr. McGilton, 

I'm pleased to inform you that your manuscript has been deemed suitable for publication in PLOS ONE. Congratulations! Your manuscript is now being handed over to our production team.

Kind regards, 

on behalf of

Dr. Mario Ulises Pérez-Zepeda 

Academic Editor

PLOS ONE